# Risk of Polypharmacy and Its Outcome in Terms of Drug Interaction in an Elderly Population: A Retrospective Cross-Sectional Study

**DOI:** 10.3390/jcm12123960

**Published:** 2023-06-10

**Authors:** Reham M. Alhumaidi, Ghazi A. Bamagous, Safaa M. Alsanosi, Hamsah S. Alqashqari, Rawabi S. Qadhi, Yosra Z. Alhindi, Nahla Ayoub, Alaa H. Falemban

**Affiliations:** 1Department of Pharmacology and Toxicology, Faculty of Medicine, Umm Al-Qura University, Makkah 24382, Saudi Arabia; pharmd_rain@hotmail.com (R.M.A.); gabamagous@uqu.edu.sa (G.A.B.); yzhindi@uqu.edu.sa (Y.Z.A.); naayoub@uqu.edu.sa (N.A.); ahfalemban@uqu.edu.sa (A.H.F.); 2Department of Community Medicine, Faculty of Medicine, Umm Al-Qura University, Makkah 24382, Saudi Arabia; hsqashqri@uqu.edu.sa; 3Institute of Cardiovascular and Metabolic Health, University of Glasgow, Glasgow G12 8QQ, UK; rsqadhi@uqu.edu.sa

**Keywords:** polypharmacy, elderly, drug interaction

## Abstract

The simultaneous use of multiple drugs—termed ‘polypharmacy’—is often required to manage multiple physiological and biological changes and the interplay between chronic disorders that are expected to increase in association with ageing. However, by increasing the number of medications consumed, the risk of undesirable medication reactions and drug interactions also increases exponentially. Hence, knowledge of the prevalence of polypharmacy and the risk of potentially serious drug–drug interactions (DDIs) in elderly patients should be considered a key topic of interest for public health and health care professionals. **Methods:** Prescription and demographic data were collected from the electronic files of patients who were aged ≥ 65 years and attended Al-Noor Hospital in Makkah, Saudi Arabia, between 2015 and 2022. The Lexicomp^®^ electronic DDI-checking platform was used to evaluate the patients’ medication regimens for any potential drug interactions. **Results:** A total of 259 patients were included in the study. The prevalence of polypharmacy among the cohort was 97.2%: 16 (6.2%) had minor polypharmacy, 35 (13.5%) had moderate polypharmacy, and 201 (77.6%) had major polypharmacy. Of the 259 patients who were taking two or more medications simultaneously, 221 (85.3%) had at least one potential DDI (pDDI). The most frequently reported pDDI under category X that should be avoided was the interaction between clopidogrel and esomeprazole and was found in 23 patients (18%). The most frequently reported pDDI under category D that required therapeutic modification was the interaction between enoxaparin and aspirin, which was found in 28 patients (12%). **Conclusions:** It is often necessary for elderly patients to take several medications simultaneously to manage chronic diseases. Clinicians should distinguish between suitable, appropriate and unsuitable, inappropriate polypharmacy, and this criterion should be closely examined when establishing a therapeutic plan.

## 1. Introduction

Older adults (>65 years) are a growing patient population. According to the World Health Organization, the proportion of elderly persons who are aged above 65 years old in the global population is increasing at a pace that is more than three times that of other age groups [1]. Accordingly, most of the population is expected to undergo or have already undergone the natural ageing process and experienced multiple physiological and biological changes that impact their bodies and quality of life. Frailty, functional limitations, falls, depression, cognitive impairment, incontinence, and malnutrition are among the most common health problems that interfere with elderly individuals’ day-to-day living [2]. Owing to these combined health conditions and the interaction of chronic disorders, which are also expected to increase in association with ageing, multiple drugs are often used simultaneously—termed ‘polypharmacy’—to manage elderly individuals’ health. An analysis of a random sample of patients aged 50–80 years found that around 37% of patients who were suffering from more than three chronic diseases took five or more medications per day, of whom 44% took at least one potentially inappropriate medication [3]. Another study found that people over the age of 65 typically spend around USD 3 billion on procuring drugs each year, accounting for one-third of all prescriptions and 40% of non-prescription medications [4].

The term ‘polypharmacy’ simply denotes the use of multiple medications by a patient. It may constitute the concurrent use of two or more [5], four or more [6], or five or more distinct prescription medications [3,7,8]. Alternatively, it may denote the use of two or more medications for 240 days or longer [9]; the daily use of multiple prescriptions, including high-risk medications; or an individual’s non-essential use of multiple medications [10]. However, a systemic review assessing the definition of polypharmacy suggested that it may be necessary to move from numerical definitions to the term ‘appropriate polypharmacy’. In a comorbidity-based context, such a definition could improve the assessment of the use of multiple medications [11]. If multiple medications are administered inappropriately, the term ‘polypragmasy’ is used [12].

Irrespective of the term’s precise definition, unintended negative consequences of the use of multiple medications are a major cause for concern. Increasing the number of medications consumed increases the risk of undesirable reactions exponentially; with two drugs, the probability of an adverse drug reaction (ADR) is 13%, which rises to 58% with five medications and to 82% when seven or more medications are used [13].

Just as the risk of an ADR increases in tandem with the number of medications consumed, the risk of drug–drug interactions (DDIs) also increases. DDIs are expected harmful reactions to two or more pharmacological agents taken concurrently and may be attributed to pharmacokinetic (absorption, distribution, metabolism, and clearance) or pharmacodynamic (when two or more pharmacological agents affect the site of action) changes. The physiological changes that occur in association with ageing, when combined with polypharmacy, increase the risk of DDIs.

A study investigating the problem of drug interactions in an emergency department found that drug interactions were present in 16% of the subjects, with an incidence rate ranging from 5.6% for those taking two drugs to 100% for those taking seven drugs. A total of 20% of the drug interactions appeared to have potential clinical significance [14]. A systematic review identified DDIs as a significant cause of 22.2% and 8.9% of hospital admissions and hospital visits, respectively [15].

In light of the above findings, the prevalence of polypharmacy and the risk of potentially serious DDIs in elderly patients should be acknowledged as a key public health concern and a topic of interest for health care professionals. The present study’s aim was to determine the risk of polypharmacy among elderly patients through evaluating the prevalence and severity of subsequent potentially serious DDIs at Al-Noor Hospital in Makkah, Saudi Arabia.

## 2. Materials and Methods

This study is an observational, retrospective cross-sectional study conducted at Al-Noor Hospital, Makkah City, Kingdom of Saudi Arabia, and was approved by the Ethics Committee of the Institutional Review Board, Ministry of Health, Saudi Arabia. The approval number is (H-02-K-076-1021-581).

Al-Noor Hospital is a specialist, secondary-level hospital serving a population of 364,255 inhabitants, with 565 beds and six different specialised medical centres.

The sample size was calculated using a 95% confidence interval, a 5% margin of error, and an estimated sample of 218 patients. The inclusion criteria consisted of male and female patients, aged ≥65 years, who contacted any medical unit at Al-Noor Hospital from 2015 to 2022. Patients who had incomplete data were excluded. Data pertaining to age, sex, laboratory results, and medication lists were collated from the patients’ medical files. All medications listed in one single prescription were included, except for those given in one shot or over a course of less than three days.

### 2.1. Defining Polypharmacy

We analysed the number of different medications dispensed in a single prescription, including all medications with the exception of topical drugs and ionotropic drugs used in the intensive care unit. Based on the prescriptions, we evaluated the degree of polypharmacy, which was further categorised into no polypharmacy (less than two medications), minor polypharmacy (two to three medications), moderate polypharmacy (four to five medications), and major polypharmacy (more than five medications) [16].

### 2.2. Defining Potentially Serious DDIs

The Lexicomp^®^ electronic DDI-checking platform was used to evaluate patient medication regimens for potential DDIs (pDDIs) [15]. This software identifies and classifies pDDIs into five types according to their degree of clinical significance. The Lexicomp^®^ (Wolters Kluwer Health Inc. Riverwoods, IL, USA) database system provides accurate information concerning each pDDI’s risk, type, mechanism, and distribution pattern. It also provides recommendations regarding how DDIs might be prevented or managed if they occur. The DDI types are classified as follows: type A—no known interaction; type B—minor or mild interaction; type C—moderate or significant interaction; type D—severe or significant interaction; and type X—contraindication or avoid combination. We focused exclusively on clinically relevant DDIs, i.e., types D and X, which are associated with significant risk and contraindication and are highly likely to cause severe adverse events that alter the effectiveness or toxicity of one or more medications.

### 2.3. Statistical Methods

Statistical analysis was conducted using SPSS 22nd edition. Numeric variables are presented as mean, standard deviation, and range. Categorical variables are presented as frequency and percentages. Pearson’s correlation test was used to correlate the number of multiple drugs used, various DDIs, and age. Any *p*-value < 0.05 was considered significant.

## 3. Results

The data were obtained from 259 patients who were older than 65 years, with 103 (39.8%) females and 156 (60.2%) males. The mean age of this study population was 76.2 ± 7.9 years (Table 1).

The average number of comorbidities in this cohort was 3.4 ± 1.5. The average number of drugs used in the cohort was 8 ± 4. The participants showed a high prevalence of hypertension and ischaemic heart disease (57.5% and 54.8%, respectively).

Five patients were taking less than two drugs, and thus, the prevalence of polypharmacy among this cohort was 98%, with 16 (5.1%) having minor polypharmacy, 32 (10%) having moderate polypharmacy, and 263 (83%) having major polypharmacy (Table 2).

Of the 259 patients taking two or more medications, 221 (85.3%) had at least one pDDI (Table 3). The drugs most commonly implicated in type D pDDIs were aspirin, enoxaparin, and clopidogrel (Table 4).

The interaction between aspirin and enoxaparin was the combination most commonly involved in major type D potential drug interactions, while the interaction between clopidogrel and esomeprazole was the most commonly implicated in type X potential drug interactions. The prevalence of polypharmacy was high at an advanced age; however, the association between polypharmacy and age was statistically insignificant (chi-square: χ^2^ = 1.291, *p* = 0.524) (Table 5). According to the chi-square test of independence, no statistically significant association between sex and polypharmacy was observed (χ^2^ = 0.287, *p* = 0.592). The variables associated with an increased risk of pDDIs were polypharmacy and the presence of comorbidities.

## 4. Discussion

It is unsurprising that polypharmacy is a key cause for concern in elderly patients as their age increases the probability of becoming ill. Various chronic diseases, as well as geriatric syndromes, require medications to manage their symptoms. In an meta-analysis of data pooled from 26 studies, the prevalence of polypharmacy was higher in studies associated with a population aged ≥65 years (45 studies, 95% CI: 37 to 54%) than in studies with a population aged <65 years (25 studies, 95% CI: 15–35%, *p* < 0.01) [16]. In a nationwide observational study that examined over 2 billion older Americans’ medical visits between 2009 and 2016, polypharmacy was documented in 65.1% of the cases, and patients with major polypharmacy tended to be older than those with moderate or minor polypharmacy [16]. In our study, 97% of the patients had taken more than two medications, among whom major polypharmacy was encountered in 79% of the patients. As expected, the number of medications increased as the number of comorbidities increased, although the analysis did not show a significant difference between the polypharmacy categories regarding the number of comorbidities. Even so, age and gender had no significant effects on the categories of polypharmacy. 

Indeed, what is surprising is that the prevalence of polypharmacy in our study exceeded what had been reported in previous studies. However, the results of this study were based on medication data collected from one prescription in the presence of the corresponding disease for both in- and outpatients, regardless of the health care unit attended. In the abovementioned American observational study, the researchers collected data from outpatient visits only, which might have resulted in an underestimation of prescriptions among the elderly patients included in the study. Indeed, several studies had observed a difference in the prevalence of polypharmacy across health care settings, with outpatient settings having a lower rate (23–37%) and inpatient hospital settings having a higher rate (52–73%) [17,18,19,20].

The high prevalence of polypharmacy observed in our study should be interpreted with caution as confounding by indication could not be excluded. The majority of our patients had more than three chronic diseases (71.8%), and major polypharmacy was most significant in patients with ischaemic heart disease and cerebrovascular events (*p*-value of 0.004 and 0.038, respectively). Given that it is expected that at least one medication would be prescribed to treat each disease, these patients would inevitably be subjected to polypharmacy. Most publications on this topic have not provided a measure of comorbidities that may influence interpretations.

It should further be noted that we chose the threshold of two medications to be considered polypharmacy, which might theoretically overestimate the prevalence. Indeed, no consensus has been established regarding the number of medications required for the definition of polypharmacy, with the threshold ranging between 2 and 21. A threshold of ≥2 medications was applied in an American observational study and in 37% of the studies included in a meta-analysis [16,18]. Nonetheless, studies applying a threshold of ≥2 medications reported a lower prevalence of polypharmacy (22 studies, 95% CI: 10–35%) than those that applied a threshold of ≥5 medications (40 studies, 95% CI 0–47%; *p** 0.01), which appears to be somewhat counterintuitive [18].

The term ‘hyper-polypharmacy’ has emerged in several studies to denote the concomitant use of ≥10 medications. As noted in a meta-analysis investigating the prevalence of polypharmacy in older adults in India, the prevalence of hyper-polypharmacy in developed countries was 1% in the USA, 2.1% in New Zealand, 8% in Australia, 18% in Sweden, and 28% in Finland [19]. In the meta-analysis itself, the prevalence in India was 31% [19]. Our study identified 105 (40%) patients with hyper-polypharmacy. 

Regarding the polypharmacy outcomes in the present study, DDIs emerged as a key aspect. It appeared that almost 85% of the patients in this cohort were at risk for DDIs, whereby the reported polypharmacy led to an average of seven pDDIs, with approximately one-third of the patients exposed to a potential type X DDI. 

However, it should be borne in mind that we calculated the prevalence of DDIs based on a DDI checker rather than the actual reports of clinicians or patients. We, therefore, consider all reported DDIs to be potential, that is, these DDIs were predicted but not reported or registered. Furthermore, it is worth mentioning that not all DDIs identified by the DDI checker were clinically significant and warranting avoidance, even in the case of severe interactions. As such, clinicians should judge DDIs by factoring in their patients’ needs, rather than relying on the data obtained from a checker. 

Although most pDDIs reported in this study were of a ‘C’ risk rating with moderate severity, considerably more attention is warranted regarding the higher-severity classifications of DDIs—types D and X. In our study, the most frequently reported pDDI under category X that should be avoided was the interaction between clopidogrel and esomeprazole and was observed in 23 patients (18%) with the risk of therapy failure and loss of clopidogrel’s protective cardiovascular benefits. This interaction is pharmacokinetic since esomeprazole inhibits the activity of CYP2C19, an enzyme involved in clopidogrel’s metabolism, resulting in a decreased level of clopidogrel’s active metabolite [17]. Replacement of esomeprazole with rabeprazole or pantoprazole is, thus, highly recommended. Based on our study of this pDDI, we did not collect any information as to whether the patients experienced any cardiovascular consequences as a result of this combination.

Regarding type D, which requires therapy modification, the pharmacodynamic interaction between enoxaparin and aspirin was the most frequently observed and was found in 28 patients (12%). Given that the risk of bleeding increases with both antiplatelet agents and enoxaparin, it is recommended that antiplatelet agents be discontinued prior to initiating enoxaparin where possible [17,18]. Notably, this combination could be concomitant with other combinations or interactions encountered in a single patient. In other words, a given patient may experience an interaction between enoxaparin and aspirin at the same time as another combination of drugs that may cause bleeding; thus, the chances of bleeding increase substantially, particularly for this population, for whom bleeding is already a risk [20,21,22]. However, a combination of antiplatelet agents and enoxaparin may be appropriate for some patients suffering from ischaemic heart disease and atrial fibrillation [23]. Whether such combinations have been deemed inappropriate for our patients or whether there is a solid clinical reason to keep these patients on this combination is unclear. Moreover, the question of whether any DDIs and, by extension, polypharmacy are considered inappropriate and must be avoided is of grave concern and warrants meticulous clinical reasoning and assessment.

## 5. Conclusions

Several medications are usually necessary to control chronic diseases. Therefore, it would be unfair to judge the use of multiple medications as wholly inappropriate. Clinicians should distinguish between suitable, appropriate and non-suitable, inappropriate polypharmacy, and this criterion must be closely examined during the process of establishing therapeutic plans for elderly patients. While the task is challenging, it would likely be made more convenient if clinicians implement such considerations within their routine daily practices; the number of drugs used by older adults should always be minimised, and nonpharmacological treatment options must be considered. While it is not necessary to restrict all patients to just a single medication, it is important to confirm that the combination of medications prescribed to patients is actually warranted and in no way harmful.

## Figures and Tables

**Table 1 jcm-12-03960-t001:** Demographics and frequency of comorbidities.

Characteristics	N (%) or Mean ± SD
Female sex	129 (40.8%)
Mean age (years)	76.4 ± 7.9
Comorbidities	3.2 (1.4%)
IHD	185 (58.5%)
HTN	183 (57.9%)
DM	142 (44.9%)
Infections	80 (25.3)
Heart Failure	53 (16.8%)
COVID-19	49 (15.5%)
Renal Insufficiency	35 (11.1%)
Cerebrovascular events	30 (9.5%)
Dementia	8 (2.5%)

IHD = Ischemic heart diseases; HTN = Hypertension; DM = Diabetes Mellitus.

**Table 2 jcm-12-03960-t002:** Baseline characteristics according to degree of polypharmacy.

Characteristics	No Polypharmacy (<2 Medications)	Minor Polypharmacy (2–3 Medications)	Moderate Polypharmacy (3–4 Medications)	Major Polypharmacy (>5 Medications)	*p*-Value
No. of patients	5 (1.6%)	16 (5.1%)	32 (10.1%)	263 (83.2%)	
Age (year), median (IQR)	75 (8.5)	76 (14)	79 (15.25)	75 (11)	0.283
Female sex, N (%)	1 (0.4%)	15 (5.8%)	5 (1.9%)	81 (31.3%)	0.549
Comorbidities, median (IQR)	3 (1)	3 (1.5)	3 (1.5%)	3 (2)	0.429

IQR = Interquartile range.

**Table 3 jcm-12-03960-t003:** Medication usage and frequency of potential drug–drug interactions.

Characteristics	N (%) or Mean ± SD
Number of prescribed drugs in each patient, median (IQP)	7 ± 2.0
Number of potential interactions in each patient	6.9 ± 7.0
Number of type C interactions	214 (96%)
Number of type D interactions	97 (44%)
Number of type X interactions	72 (32.6%)

**Table 4 jcm-12-03960-t004:** Medications incorporated in type D DDIs among the included patients.

**Drug Interaction Type D**		**N (%)**
Enoxaparin	Aspirin	28 (12%)
Heparin	Aspirin	20 (8.0%)
Heparin	Clopidogrel	19 (8.0%)
Drug interaction Type X		N (%)
Clopidogrel	Esomeprazole	23 (18%)
Cefuroxime	Omeprazole	8 (6.0%)
Tiotropium	Ipratropium	8 (6.0%)

DDI = drug-drug interactions.

**Table 5 jcm-12-03960-t005:** Correlation matrix between DDIs, number of diagnoses, number of drugs, and age.

r (*p*-Value)	Age	No. of Diagnose	No. of Drugs	No. of DDI	Type C	Type D	Type X
Age		−0.112 (0.072)	0.001 (0.987)	0.012 (0.85)	0.062 (0.367)	−0.11 (0.283)	−0.189 (0.112)
No. of diagnose			0.257 (0.0001)	0.184 (0.003)	0.133 (0.052)	0.06 (90.502)	−0.057 (0.633)
No. of drugs				0.772 (0.0001)	0.676 (0.0001)	0.235 (0.021)	0.373 (0.001)
No. of DDI					0.967 (0.0001)	0.339 (0.001)	0.290 (0.013)
Type C						0.19 (0.069)	0.149 (0.225)
Type D							0.103 (0.55)
Type X							

DDIs = drug-drug interactions.

## Data Availability

Not applicable.

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
