# Peer review of "Risk of Polypharmacy and Its Outcome in Terms of Drug Interaction in an Elderly Population: A Retrospective Cross-Sectional Study"

_jcm, 2023, doi:10.3390/jcm12123960_

Round 1

Reviewer 1 Report

1. Author did not address the Inclusion/Exclusion criteria well,e.g. whether Pregnant and lactating women have been taken, 

2. Why did the Pt's of age more than  65 yrs were included, why not from  60 yrs. of age? . 

3. In the Exclusion criteria Author did not specifically mention whether supplements (e.g. multivitamins ,Caclcium preperations) ,or specific chemotherapy (antiretrovirals and anticancer drugs) are excluded from polypharmacy criteria.

4. Author did not use a WHO/INRUD prescriber & detailed encounters proforma data collection sheet for the analysis of core medicine use indicators such as polypharmacy in his study

5. In data analysis for more sensitivity and specificity for checking drug-drug interactions, the Epocrates online and Medscape drug reference database may be used apart from Lexicomp electronic DDI -checking platform ( Ref. 

Polypharmacy & the occurrence of potential drug-drug interaction-------in Durban ,South Africa  . Bojuwoye A O et al , J. Pharm Policy Pract 2022;15:1)

Author Response

Thank you for your valuable feedback

  1. Author did not address the Inclusion/Exclusion criteria well,e.g. whether Pregnant and lactating women have been taken, 

The study was intended for geriatric patients, so finding pregnant or lactating women among the participants was unexpected.

  1. Why did the Pt's of age more than 65 yrs were included, why not from  60 yrs. of age?

There is indeed no general agreement on the age at which a person becomes old. However, medical research often defines a person as elderly when they are 65 years of age or above (Sabharwal, Wilson et al. 2015).  Saudi medical sections along with most studies conducted in the area used an age of 65 years as a definition of 'elderly' or older person (https://www.moh.gov.sa/en/Ministry/Information-and-services/Pages/Elderly.aspx), (Salam 2023). Therefore, we chose to adopt the same approach, including patients of age more than 65 years.

  1. In the Exclusion criteria Author did not specifically mention whether supplements (e.g. multivitamins ,Caclcium preperations) ,or specific chemotherapy (antiretrovirals and anticancer drugs) are excluded from polypharmacy criteria.

There was no specific medication class that is excluded. All medications listed in a one single prescription are included except those given in one shot or in a course of less than three days.

  1. Author did not use a WHO/INRUD prescriber & detailed encounters proforma data collection sheet for the analysis of core medicine use indicators such as polypharmacy in his study

Prescribing indicators are useful for assessing the prescribing practice of the healthcare delivery system. Our study aim was beyond assessing prescribing practice, rather the aim was to determine the risk of polypharmacy among geriatric patients, evaluating the prevalence and severity of subsequent potentially serious DDIs

  1. In data analysis for more sensitivity and specificity for checking drug-drug interactions, the Epocrates online and Medscape drug reference database may be used apart from Lexicomp electronic DDI -checking platform ( Polypharmacy & the occurrence of potential drug-drug interaction-------in Durban ,South Africa  . Bojuwoye A O et al , J. Pharm Policy Pract 2022;15:1)

Lexicomp® electronic DDI-checking was chosen to assess DDI, since it is more familiar to Saudi Arabian healthcare providers. Most hospitals include this platform in their electronic medical decision support tools.

Reviewer 2 Report

The paper, based on retrospective and monocentric study, provides some useful information however of replicative nature.

Introduction

If the patient files analyzed were from subjects over 65 years, you should use the term “elderly” instead “geriatric”; in the title and throughout the text. Age of 65 is the most common definition at which a person is considered to be elderly, including World Health Organization.

In the introduction polypharmacy should be presented against “polypragmasy” (inappropriate administration of multiple medications).’

Definition of polypharmacy is more complex, and nicely revised in Masnoon, N., Shakib, S., Kalisch-Ellett, L. et al. What is polypharmacy? A systematic review of definitions. BMC Geriatr 17, 230 (2017). https://doi.org/10.1186/s12877-017-0621-2; and should be presented in the introduction.

Results

Do you have data from specific clinical departments (internal medicine, surgery, gynecology and obstetrics)

Did you observe in patients files clinical consequences of DDI classified as D or X (or just risks were only according to the classification system used). Those cases should be analyzed and the results provided. This builds the knowledge about DDIs.

Discussion

Another D class interactions reported - between cefuroxime and omeprazole as well ipratropium and tiotropium are not discussed.

Author Response

Thank you for your valuable feedback.

The paper, based on retrospective and monocentric study, provides some useful information however of replicative nature.

Introduction

If the patient files analyzed were from subjects over 65 years, you should use the term “elderly” instead “geriatric”; in the title and throughout the text. Age of 65 is the most common definition at which a person is considered to be elderly, including World Health Organization.  Done

In the introduction polypharmacy should be presented against “polypragmasy” (inappropriate administration of multiple medications).’ Done

Definition of polypharmacy is more complex, and nicely revised in Masnoon, N., Shakib, S., Kalisch-Ellett, L. et al. What is polypharmacy? A systematic review of definitions. BMC Geriatr 17, 230 (2017). https://doi.org/10.1186/s12877-017-0621-2; and should be presented in the introduction. Done

Results

Do you have data from specific clinical departments (internal medicine, surgery, gynecology and obstetrics)

We do not have this information. Identifying which departments or care units are at risk of polypharmacy would have been beneficial, but unfortunately, we did not register this information during data collection. Cases of any patients male or female aged 65 and above who contacted any medical care unit including medicine, surgery , ER and ICU of Al-Noor hospital have been included. Since the study was intended for geriatric patients, no cases have been taken from gynecology and obstetrics department.

Did you observe in patients’ files clinical consequences of DDI classified as D or X (or just risks were only according to the classification system used). Those cases should be analyzed and the results provided. This builds the knowledge about DDIs.

We calculated the prevalence of DDI based on the DDI checker rather than the actual reports of clinicians or patients. We therefore consider all reported DDIs to be potential—that is, which is predicted but not reported or registered. Next study is planned to be conducted investigating the prevalence of any adverse effect as consequence of DDI. 

Discussion

Another D class interactions reported - between cefuroxime and omeprazole as well ipratropium and tiotropium are not discussed.

We chose to discuss the one most frequent pDDI.

Reviewer 3 Report

This is a very interesting and well-developed work, based on a topic that has been little explored in recent times despite its high frequency and great population impact. The title is concrete and well descriptive. The abstract is appropriately divided representing the different sections of the article, and presenting the main results findings with correctly expressed numerical values. The introduction is a bit long, although it is of high quality and appropriately describes all the concepts (even some that are somewhat excessive) for the understanding of the subject, setting out the objectives of the work correctly. The materials and methods are adequate and well described, ensuring the reproducibility of the work. The statistical procedures are correct. The results are clear, concrete and well expressed. They are used very well constructed that allow to properly interpret the result. The discussion is correct and extensive. It is compared with other similar studies, and strengths and weaknesses are described. For all that has been said, I believe that it is a high-quality work that is in a position.

None

Author Response

It is our pleasure to hear your valuable feedback.

Thank you.